# Episodic Memory for Learning Subjective-Timescale Models

## Abstract

In model-based learning, an agent's model is commonly defined over transitions between consecutive states of an environment even though planning often requires reasoning over multi-step timescales, with intermediate states either unnecessary, or worse, accumulating prediction error. In contrast, intelligent behaviour in biological organisms is characterised by the ability to plan over varying temporal scales depending on the context. Inspired by the recent works on human time perception, we devise a novel approach to learning a transition dynamics model, based on the sequences of episodic memories that define the agent's *subjective timescale* – over which it learns world dynamics and over which future planning is performed. We implement this in the framework of active inference and demonstrate that the resulting subjective-timescale model (STM) can systematically vary the *temporal extent* of its predictions while preserving the same computational efficiency. Additionally, we show that STM predictions are more likely to introduce future salient events (for example new objects coming into view), incentivising exploration of new areas of the environment. As a result, STM produces more informative action-conditioned roll-outs that assist the agent in making better decisions. We validate significant improvement in our STM agent's performance in the Animal-AI environment against a baseline system, trained using the environment's objective-timescale dynamics.

## 1 Introduction

An agent endowed with a model of its environment has the ability to predict the consequences of its actions and perform planning into the future before deciding on its next move. Models can allow agents to simulate the possible action-conditioned futures from their current state, even if the state was never visited during learning. As a result, *model-based* approaches can provide agents with better generalization abilities across both states and tasks in an environment, compared to their *model-free* counterparts (Racanière et al., 2017; Mishra et al., 2017).

The most popular framework for developing agents with internal models is model-based reinforcement learning (RL). Model-based RL has seen great progress in recent years, with a number of proposed architectures attempting to improve both the quality and the usage of these models (Kaiser et al., 2020; Racanière et al., 2017; Kansky et al., 2017; Hamrick, 2019). Nevertheless, learning internal models affords a number of unsolved problems. The central one of them is model-bias, in which the imperfections of the learned model result in unwanted over-optimism and sequential error accumulation for long-term predictions (Deisenroth & Rasmussen, 2011). Long-term predictions are additionally computationally expensive in environments with slow temporal dynamics, given that all intermediary states must be predicted. Moreover, slow world dynamics[1] can inhibit the learning of dependencies between temporally-distant events, which can be crucial for environments with sparse rewards. Finally, the temporal extent of future predictions is limited to the *objective* timescale of the environment over which the transition dynamics has been learned. This leaves little room for flexible and context-dependent planning over varying timescales which is characteristic to animals and humans (Clayton et al., 2003; Cheke & Clayton, 2011; Buhusi & Meck, 2005).

The final issue exemplifies the disadvantage of the classical view on internal models, in which they are considered to capture the ground-truth transition dynamics of the environment. Furthermore,

---

[1]Worlds with small change in state given an action

in more complex environments with first-person observations, this perspective does not take into account the apparent subjectivity of first-person experiences. In particular, the agent's learned representations of the environment's transition dynamics implicitly include information about *time*. Little work has been done to address the concept of time perception in model-based agents (Deverett et al., 2019). Empirical evidence from the studies of human and animal cognition suggests that intelligent biological organisms do not perceive time precisely and do not possess an explicit clock mechanism responsible for keeping track of time (Roseboom et al., 2019; Sherman et al., 2020; Hills, 2003). For instance, humans tend to perceive time slower in environments rich in perceptual content (e.g. busy city), and faster in environments with little perceptual change (e.g. empty field). The mechanisms of subjective time perception still remain unknown; however, recent computational models based on episodic memory were able to closely model the deviations of human time perception from veridical perception (Fountas et al., 2020b).

Inspired by this account, in this work we propose *subjective-timescale model* (STM), an alternative approach to learning a transition dynamics model, by replacing the objective timescale with a subjective one. The latter represents the timescale by which an agent perceives events in an environment, predicts future states, and which is defined by the sequences of episodic memories. These memories are accumulated on the basis of saliency (i.e. how poorly an event was predicted by the agent's transition model), which attempts to mimic the way humans perceive time, and resulting in the agent's ability to plan over varying timescales and construct novel future scenarios.

We employ active inference as the agent's underlying cognitive framework. Active inference is an emerging framework within computational neuroscience, which attempts to unify perception and action under the single objective of minimising the free-energy functional. Similar to model-based RL, an active inference agent relies almost entirely on the characteristics and the quality of its internal model to make decisions. Thus, it is naturally susceptible to the previously mentioned problems associated with imperfect, objective-timescale models. The selection of active inference for the purposes of this paper is motivated by its biological plausibility as a normative framework for understanding intelligent behaviour (Friston et al., 2017a; 2006), which is in line with the general theme of this work. Furthermore, being rooted in variational inference, the free energy objective generates a distinct separation between the information-theoretic quantities that correspond to the different components of the agent's model, which is crucial to define the memory formation criterion.

We demonstrate that the resulting characteristics of STM allow the agent to automatically perform both short- and long-term planning using the same computational resources and without any explicit mechanism for adjusting the temporal extent of its predictions. Furthermore, for long-term predictions STM systematically performs temporal jumps (skipping intermediary steps), thus providing more informative future predictions and reducing the detrimental effects of one-step prediction error accumulation. Lastly, being trained on salient events, STM much more frequently imagines futures that contain epistemically-surprising events, which incentivises exploratory behaviour.

## 2    RELATED WORK

**Model-based RL.** Internal models are extensively studied in the field of model-based RL. Using linear models to explicitly model transition dynamics has achieved impressive results in robotics (Levine & Abbeel, 2014a; Watter et al., 2015; Bagnell & Schneider, 2001; Abbeel et al., 2006; Levine & Abbeel, 2014b; Levine et al., 2016; Kumar et al., 2016). In general, however, their application is limited to low-dimensional domains and relatively simple environment dynamics. Similarly, Gaussian Processes (GPs) have been used (Deisenroth & Rasmussen, 2011; Ko et al., 2007). Their probabilistic nature allows for state uncertainty estimation, which can be incorporated in the planning module to make more cautious predictions; however, GPs struggle to scale to high-dimensional data. An alternative and recently more prevalent method for parametrising transition models is to use neural networks. These are particularly attractive due to their recent proven success in a variety of domains, including deep model-free RL (Silver et al., 2017), ability to deal with high-dimensional data, and existence of methods for uncertainty quantification (Blundell et al., 2015; Gal & Ghahramani, 2016). Different deep learning architectures have been utilised including fully-connected neural networks (Nagabandi et al., 2018; Feinberg et al., 2018; Kurutach et al., 2018) and autoregressive models (Ha & Schmidhuber, 2018; Racanière et al., 2017; Ke et al., 2019), showing promising results in environments with relatively high-dimensional state spaces. In particular, autoregressive

architectures, such as Long Short-Term Memory (LSTM) (Hochreiter & Schmidhuber, 1997), are capable of modelling non-Markovian environments and of learning temporal dependencies. Nevertheless, LSTMs are still limited in their ability to learn relations between temporally-distant events, which is exacerbated in environments where little change occurs given an action.

Uncertainty quantification using ensemble methods (Kalweit & Boedecker, 2017; Clavera et al., 2020; Buckman et al., 2018) or Bayesian neural networks (McAllister & Rasmussen, 2016; Depeweg et al., 2017) have been proposed to tackle model bias and sequential error accumulation. Other works have focused on techniques to create more accurate long-term predictions. Mishra et al. (2017) used a segment-based approach to predict entire trajectories at once in an attempt to avoid one-step prediction error accumulation. A work by Ke et al. (2019) used an autoregressive model and introduced a regularising auxiliary cost with respect to the encodings of future observations, thus forcing the latent states to carry useful information for long-horizon predictions. In contrast, the work presented in this paper re-focuses the objective from attempting to create better parametrisation techniques or mitigating methods to simply transforming the timescale over which the dynamics of an environment is learned. As will be seen, our approach can lead to more accurate and efficient long-term predictions without compromising agent's ability to plan over short time-horizons.

**Episodic Memory.** In neuroscience, episodic memory is used to describe autobiographical memories that link a collection of first-person sensory experiences at a specific time and place (Tulving, 1972). Past studies in the field suggest that episodic memory plays an important role in human learning (Mahr & Csibra, 2017), and may capture a wide range of potential functional purposes, such as construction of novel future scenarios (Schacter et al., 2007; 2012; Hassabis et al., 2007), mental time-travel (Michaelian, 2016) or assisting in the formation of new semantic memories (Greenberg & Verfaellie, 2010). A recent computational model of episodic memory (Fountas et al., 2020b) also relates it to the human ability to estimate time durations.

The application of episodic memory in reinforcement learning has been somewhat limited. Some works have employed simple forms of memory to improve the performance of a deep model-free RL agent via experience replay (Mnih et al., 2015; Espeholt et al., 2018; Schaul et al., 2016). However, these methods do not incorporate information about associative or temporal dependencies between the memories (Hansen et al., 2018). Read-write memory banks have also been implemented alongside gradient-based systems (memory-augmented neural networks) for assisting in learning and prediction (Graves et al., 2014; 2016; Hung et al., 2019; Oh et al., 2016; Jung et al., 2018). Further, episodic memory has been used for non-parametric Q-function approximation (Blundell et al., 2016; Pritzel et al., 2017; Hansen et al., 2018; Zhu et al., 2020). It has also been proposed to be used directly for control as a faster and more efficient alternative to model-based and model-free approaches in RL, such as instance-based control (Lengyel & Dayan, 2007; Botvinick et al., 2019; Gershman & Daw, 2017) and one-shot learning (Kaiser et al., 2017). In contrast, our paper considers a novel way of using episodic memories – in defining the agent's subjective timescale of the environment and training a transition dynamics model over the sequences of these memories.

**Active Inference.** Until now, most of the work on active inference has been done in low-dimensional and discrete state spaces (Friston et al., 2015; 2017b;c;d). Recently, however, there has been a rising interest in scaling active inference and applying it to environments with continuous and/or large state spaces (Fountas et al., 2020a; Tschantz et al., 2019; Çatal et al., 2019; Millidge, 2019; Ueltzhöffer, 2018). Although these works used deep learning techniques, their generative models have so far been designed to be Markovian and trained over the objective timescale of the environment.

## 3 BASELINE ARCHITECTURE

We take the deep active inference system devised by Fountas et al. (2020a) as the starting point with a few architectural and operational modifications. The generative model of this baseline agent is defined as $p(o_{1:t}, s_{1:t}, a_{1:t}; \theta)$, where $s_t$ denotes latent states at time $t$, $o_t$ (visual) observations, $a_t$ actions, and $\theta = \{\theta_o, \theta_s\}$ the parameters of the model. $s_t$ is assumed to be Gaussian-distributed with a diagonal covariance, $o_t$ follows Bernoulli and $a_{1:t}$ categorical distributions. For a single time step, as illustrated in Figure 1A, this generative model includes two factors, a transition model $p(s_t|s_{t-1}, a_{t-1}; \theta_s)$ and a latent state decoder $p(o_t|s_t; \theta_o)$ parametrised by feed-forward neural net-

works with parameters $\theta_s$ and $\theta_o$, respectively. We modify the transition model from the original study to predict *the change in state*, rather than the full state[2].

The agent also possesses two inference networks, which are trained using amortized inference: a habitual network $q(a_t; \phi_a)$ and observation encoder $q(s_t; \phi_s)$ parametrised by $\phi_a$ and $\phi_s$, respectively. The habitual network acts as a model-free component of the system, learning to map inferred states directly to actions. Following Fountas et al. (2020a), the variational free energy for an arbitrary time-step $t$ is defined as:

$$F_t = - \mathbb{E}_{q(s_t)} \left[ \log p(o_t|s_t; \theta_o) \right] \tag{1a}$$

$$+ D_{\mathrm{KL}} \left[ q(s_t; \theta_s) \| p(s_t|s_{t-1}, a_{t-1}; \theta_s) \right] \tag{1b}$$

$$+ \mathbb{E}_{q(s_t)} \left[ D_{\mathrm{KL}} \left[ q(a_t; \phi_a) \| p(a_t) \right] \right] \tag{1c}$$

where $p(a) = \sum_{\pi:a_1=a} p(\pi)$ is the summed probability of all policies beginning with action $a$. All the divergence terms are computable in closed-form, given the assumption about Gaussian- and Bernoulli-distributed variables. Finally, the expected free energy (EFE) of the generative model up to some time horizon $T$ can be defined as:

$$G(\pi) = \sum_{\tau=t}^{T} G(\pi, \tau) = \sum_{\tau=t}^{T} \mathbb{E}_{\tilde{q}} \left[ \log q(s_\tau, \theta|\pi) - \log p(o_\tau, s_\tau, \theta|\pi) \right], \tag{2}$$

where $\tilde{q} = q(o_\tau, s_\tau, \theta|\pi)$ and $p(o_\tau, s_\tau, \theta|\pi) = p(o_\tau|\pi)q(s_\tau|o_\tau, \pi)p(\theta|s_\tau, o_\tau, \pi)$.

To make expression 2 computationally feasible, it is decomposed such that,

$$\begin{aligned} G(\pi, \tau) = &- \mathbb{E}_{q(\theta|\pi)q(s_\tau|\theta,\pi)q(o_\tau|s_\tau,\theta,\pi)} \left[ \log p(o_\tau|\pi) \right] \\ &+ \mathbb{E}_{q(\theta|\pi)} \left[ \mathbb{E}_{q(o_\tau|\theta,\pi)} H(s_\tau|o_\tau, \pi) - H(s_\tau|\pi) \right] \\ &+ \mathbb{E}_{q(\theta|\pi)q(s_\tau|\theta,\pi)} \left[ H(o_\tau|s_\tau, \theta, \pi) \right] \\ &- \mathbb{E}_{q(s_\tau|\pi)} \left[ H(o_\tau|s_\tau, \pi) \right], \end{aligned} \tag{3}$$

where expectations can be taken by performing sequential sampling of $\theta$, $s_\tau$ and $o_\tau$ and entropies are calculated in closed-form using standard formulas for Bernoulli and Gaussian distributions. Network parameters, $\theta$, are sampled using Monte Carlo (MC) dropout (Gal & Ghahramani, 2016).

The system also makes use of *top-down attention* mechanism by introducing variable $\omega$, which modulates uncertainty about hidden states, promoting latent state disentanglement and more efficient learning. Specifically, the latent state distribution is defined as a Gaussian such that $s \sim \mathcal{N}(\boldsymbol{s}; \boldsymbol{\mu}, \boldsymbol{\Sigma}/\omega)$, where $\boldsymbol{\mu}$ and $\boldsymbol{\Sigma}$ are the mean and the diagonal covariance, and $\omega$ is a decreasing logistic function over the divergence $D_{\mathrm{KL}} \left[ q(a; \phi_a) \| p(a) \right]$.

Finally, action selection is aided with Monte Carlo tree search (MCTS), ensuring a more efficient trajectory search. Specifically, MCTS generates a weighted tree that is used to sample policies from the current timestep, where the weights refer to the agent's estimation of the EFE given a state-action pair, $\tilde{G}(s, a)$. The nodes of the tree are predicted via the transition model, $p(s_t|s_{t-1}, a_{t-1}; \theta_s)$. At the end of the search, MCTS is used to construct the action prior, $p(a) = N(a_i, s) / \sum_j N(a_j, s)$, where $N(s, a)$ is the number of times action $a$ has been taken from state $s$.

The baseline agent is trained with prioritised experience replay (PER) (Schaul et al., 2016) to mitigate the detrimental consequences of on-line learning (which was used in the original paper), and to encourage better object-centric representations. The details of the baseline implementation and training with PER can be found in Appendices B.1 and B.2, respectively.

## 4   Subjective-Timescale Model

We introduce subjective-timescale model (STM) that records sequences of episodic memories over which a new transition model is trained. As such, the system consists of a memory accumulation system to selectively record salient events, a simple action heuristic to summarise sequences of actions between memories, and an autoregressive transition model.

---

[2]This has largely become common practice in the field of model-based RL (Nagabandi et al., 2018), improving algorithm efficiency and accuracy especially in environments with slow temporal dynamics.

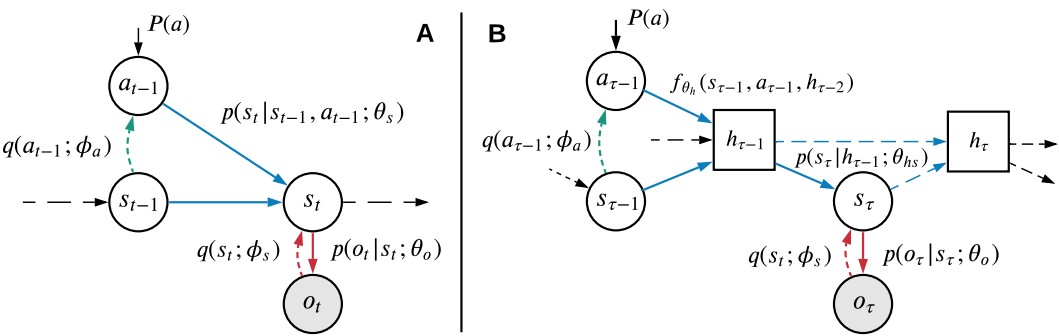

Figure 1: **A.** Baseline generative model. **B.** STM generative model with additional deterministic hidden states $h$ introduced by an LSTM.

We define a *ground-truth sequence* as a sequence of *all* states experienced in an environment during a single episode, $S_g = \{s_0, s_1, s_2, ..., s_T\}$, and an *S-sequence* (subjective sequence) as a sequence of states selectively picked by our system, and over which the new transition model would be learned, $S_e = \{s_{\tau_1}, s_{\tau_2}, s_{\tau_3}, ..., s_{\tau_N}\}$. Each unit in an S-sequence is called *an episodic memory* and consists of a set of sufficient statistics, $s = \{\boldsymbol{\mu}_s, \boldsymbol{\sigma}_s\}$, where $\boldsymbol{\mu}_s$ and $\boldsymbol{\sigma}_s$ are mean and variance vectors of a Gaussian-distributed state $s$, respectively. Additionally, each episodic memory contains a reference to its preceding (parent) episodic memory and all actions until the next one. The process of recording S-sequences is called *memory accumulation*.

## 4.1 MEMORY ACCUMULATION

Previous work on time perception and episodic memory (Fountas et al., 2020b) employed *saliency* of an event, or the generative model's prediction error, as the memory formation criterion. Selection of this criterion is informed by the experimental evidence from neuroscience on episodic memory (Greve et al., 2017; Jang et al., 2018; Rouhani et al., 2018). Inspired by this account, our memory accumulation system employs the free energy of the objective-timescale *transition model*[3] (Eq.1b) as a measure of event saliency, and forms memories when a pre-defined threshold is exceeded.

To train STM, an active inference agent moves in the environment under a pre-trained generative model described in Section 3. During this process, each transition is evaluated based on the objective transition model free energy, $D_{\mathrm{KL}}\left[q(\mathrm{s}_t; \theta_s) \| p(\mathrm{s}_t | \mathrm{s}_{t-1}, \mathrm{a}_{t-1}; \theta_s)\right]$, which represents the degree of surprise experienced by the transition model upon taking an action. If the value of the free energy exceeds a pre-defined threshold, $\epsilon$, a memory is formed and placed into an S-sequence. At the end of each episode, the recorded S-sequence is saved for later use.

We can categorise the transitions that cause higher values of transition model free energies into two main groups: epistemic surprise and model-imperfection surprise. The former refers to transitions that the model could not have predicted accurately due to the lack of information about the current state of the environment (e.g. objects coming into view). The latter refers to the main bulk of these high prediction-error transitions and stems from the inherent imperfections of the learned dynamics. Specifically, less frequently-occurring observations with richer combinatorial structure would systematically result in higher compounded transition model errors, given that these would be characteristic of more complex scenes. As will become apparent, the presence of these two categories in the recorded S-sequences results in the model's ability to vary its prediction timescale based on the perceptual context and systematically imagine future salient events.

A transition dynamics model is necessarily trained with respect to actions that an agent took to reach subsequent states. However, STM records memories over an arbitrary number of steps, thus leaving action sequences of variable length. For the purposes of this paper, we implement a simple heuristic to summarise agent's trajectories, which is enough to provide STM with the necessary information

---

[3]Components of the *total* free energy correspond to a measure of belief update for each of the networks, and therefore, loosely speaking, quantify the prediction error generated by each of the respective system constituents: autoencoder (Eqs.1a, 1b), objective-timescale transition model (Eq.1b), and habitual network (Eq.1c).

Figure 2: **STM pipeline. (A)** As the agent moves through the environment, states $s$ that exceeded a pre-defined threshold are recorded along with all successive actions $a$ in an S-sequence. **(B)** S-sequences are saved in a buffer at the end of each episode. **(C)** S-sequences are sampled for training a subjective-timescale transition model.

to learn action-conditioned predictions. We do it by estimating the angle between the agent's initial position and its final position at the time-step of the subsequent memory. Full details of this heuristic can be found in Appendix B.4.

## 4.2 TRANSITION DYNAMICS MODEL

As mentioned, S-sequences are characterised by the presence of epistemically-surprising and salient events squeezed together in the recorded episodes. As a result, training on these sequences is more conducive for learning *temporal dependencies* between important states. For this reason, we train an LSTM model over the S-sequences, which utilises internal memory states to store information about preceding inputs. In our architecture, an LSTM calculates hidden state $h_\tau$ at subjective time $\tau$ using a deterministic mapping,

$$h_\tau = f_{\theta_h}(s_\tau, a_\tau, h_{\tau-1}) = \sigma(x_\tau W_h + h_{\tau-1} U_h + b_h), \tag{4}$$

where $s_\tau$ and $a_\tau$ are the latent state and action taken at subjective time $\tau$ respectively, $x_\tau$ is the concatenated vector of $s_\tau$ and $a_\tau$, and $\theta_h = \{W_h, U_h, b_h\}$ are deterministic LSTM model parameters. Importantly, function $f_{\theta_s}$ is deterministic and serves only to encode information about preceding steps into the hidden state of the LSTM. This hidden state $h_\tau$ is then mapped to a latent state $s_{\tau+1}$ at the next subjective time $\tau + 1$ via a feed-forward neural network with random-variable parameters, $\theta_{hs}$, using $p(s_\tau|h_{\tau-1};\theta_{hs})$ with MC dropout. The parameters of both of the networks are trained via backpropagation with a loss function defined as

$$\mathcal{L} = \frac{1}{T} \sum_\tau^T D_{\mathrm{KL}}\Big[q(s_{\tau+1};\phi_s)\|p(s_{\tau+1}|f_{\theta_h}(s_\tau, a_\tau, h_{\tau-1});\theta_{hs})\Big] \tag{5}$$

The new generative model of observations is shown in Figure 1B. Because the mapping of LSTM is deterministic, the formulation of the variational free energy remains intact with the exception of the second term that now includes the state prediction produced by the network $p(s_\tau|h_{\tau-1};\theta_{hs})$ conditioned on the hidden state of the LSTM,

$$\begin{aligned} F_\tau = &- \mathbb{E}_{q(s_\tau)}\left[\log p(o_\tau|s_\tau;\theta_o)\right] \\ &+ D_{KL}\left[q(s_\tau;\phi_s)\|p(s_\tau|h_{\tau-1};\theta_{hs})\right] \\ &+ \mathbb{E}_{q(s_\tau)}\left[D_{KL}\left[q(a_\tau;\phi_a)\|p(a_\tau)\right]\right] \end{aligned} \tag{6}$$

Architectural and training details of the model can be found in Appendix B.3. The source code will be made available after the review process.

## 5 EXPERIMENTS

The Animal-AI (AAI) environment is a virtual testbed that provides an open-ended sandbox training environment for interacting with a 3D environment from first-person observations (Crosby et al.,

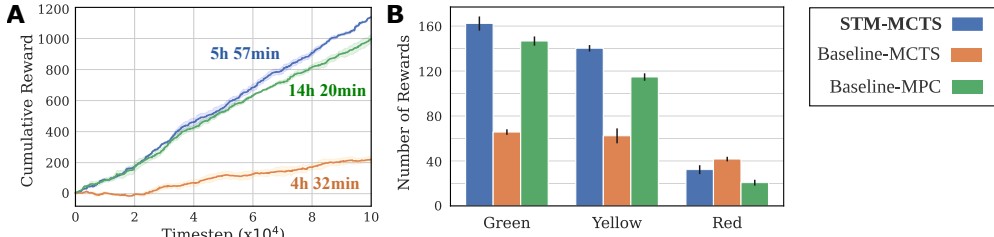

Figure 3: **Experimental Results. (A)** Cumulative rewards collected by the agents. It can be seen that the STM-MCTS agent shows improved performance even when compared with the computationally-expensive Baseline-MPC. **(B)** Mean number of rewards (spheres) by category. STM-MCTS collects more green and yellow (positive) rewards than the baseline agents. However, Baseline-MPC collects fewer (negative) red rewards, which is likely related its ability to evaluate actions after every step. Uncertainty regions and bars indicate one standard deviation over 5 runs.

2020; Crosby, 2020). In AAI, an agent is tasked with reaching a green sphere given a particular setup that may include intermediary rewards (yellow spheres), terminal negative rewards (red spheres), obstacles (e.g. walls), etc. For the purposes of this work, we use a sparsely populated configuration with single green, red, and yellow spheres, in which a successful agent would be forced to perform both short- and long-distance planning, as well as more extensive exploration of the environment.

## 5.1 EXPERIMENTAL RESULTS

We tested the STM agent using 100,000 steps in randomly-generated environments (max episode length of 500) against the baseline system with two different planning procedures – MCTS and model-predictive control (MPC). In contrast to MCTS, the MPC agent re-evaluates its plan after every action. Figure 3 summarises the experimental results. Our STM-MCTS agent outperforms the baseline systems in acquiring more rewards within the 100,000 steps. In particular, we note that the STM-MCTS agent showed significant improvement against the Baseline-MCTS. Similarly, we show that STM-MCTS model retrieves more cumulative reward than the Baseline-MPC agent, which uses a computationally expensive planning procedure. Specifically, our agent achieves a higher cumulative reward in less than half the time, ∼6 hours, compared to ∼14 hours.

## 5.2 ROLL-OUT INSPECTION

Inspecting prediction roll-outs produced by the STM-based system provides great insight into its practical benefits for the agent's performance. Specifically, our agent is capable of varying the temporal extent of its predictions and imagining future salient events.

### 5.2.1 VARYING PREDICTION TIMESCALE

Much like human perception of time changes depending on the perceptual content of the surroundings, our agent varies the prediction timescale depending on the context it finds itself in. Specifically, in the AAI environment the complexity of any given observation is primarily driven by the presence of objects, which may appear in different sizes, colours, and configurations. As a result, our agent consistently predicts farther into the future in the absence of any nearby objects, and slows its timescale, predicting at finer temporal rate, when the objects are close.

Practically, this has several important implications. First, performing temporal jumps and skipping unnecessary intermediary steps affords greater computational efficiency, and reduces the detrimental effects of sequential error accumulation, as can be seen in Figure 4. Second, while STM is able to predict far ahead, its inherent flexibility to predict over varying timescales does not compromise the agent's performance when the states of interest are close. Thus, a separate mechanism for adjusting how far into the future an agent should plan is not necessary and is implicitly handled by our model. Third, STM allows the agent to make more *informed* decisions in an environment, as it tends to populate the roll-outs with salient observations of the short- and long-term futures depending on the context. As a result, STM effectively re-focuses the central purpose of a transition model from

most accurately modeling the ground-truth dynamics of an environment to predicting states more informative with respect to the affordances of the environment.

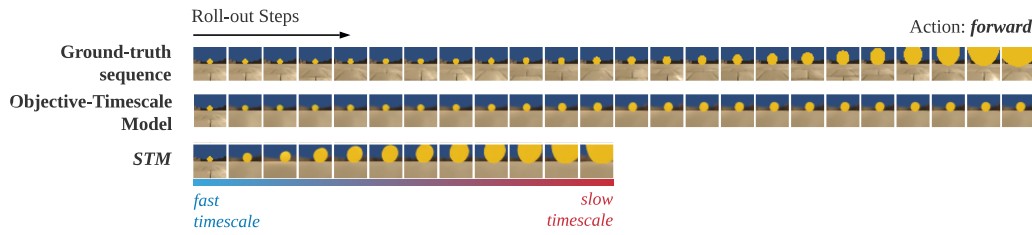

Figure 4: **STM can vary prediction timescale**, with large gaps between predictions at the start, which becomes more fine-grained as the object gets closer. Objective-timescale model suffers from slow-timescale predictions and error accumulation, resulting in poorly-informative predictions.

### 5.2.2 IMAGINING SURPRISING EVENTS

As mentioned, S-sequences frequently include epistemically-surprising transitions, which, in the context of the AAI environment, constitute events where objects come into view. As a result, STM is significantly more likely to include roll-outs with new objects appearing in the frame, in contrast to the baseline that employs the objective-timescale transition model.

The ability of the STM to imagine novel and salient future events encourages exploratory behaviour, which is distinct from the active inference agent's intrinsic exploratory motivations. We again stress that although the predicted futures may be inaccurate with respect to the ground-truth positions of the objects, they are nevertheless more *informative* with respect to the agent's *potential affordances* in the environment. This is in stark contrast with the objective-timescale model, which imagines futures in the absence of any objects. As a result, the STM agent is less prone to get stuck in a sub-optimal state, which was commonly observed in the baseline system, and is more inclined to explore the environment beyond its current position.

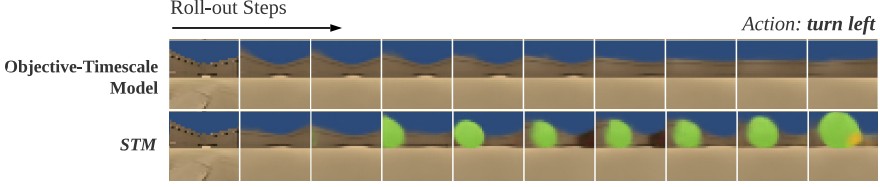

Figure 5: **STM is able to imagine surprising events.** Despite the fact that appearance of objects is a rare event in the environment, STM frequently predicts them in the roll-outs. In contrast, objective-timescale model is not capable of that as a direct corollary of its training procedure.

## 6 CONCLUSION AND FUTURE WORK

We proposed STM, a novel approach to learning a transition dynamics model with the use of sequences of episodic memories, which define an agent's more useful, subjective timescale. STM showed significant improvement against the baseline agent's performance in the AAI environment. Inspired by the problems of inaccurate and inefficient long-term predictions in model-based RL and the recent neuroscience literature on episodic memory and human time perception, we merged ideas from the different fields into one new technique of learning a forward model. We further emphasised two important characteristics of the newly-devised model – its ability to vary the temporal extent of future predictions and to predict future salient events. The application of our technique is not limited to active inference, and can be adapted for use in other model-based frameworks.

Future work may explore more generalised approaches of action summarisation and dynamic thresholding for memory formation. Another enticing direction of research is to investigate the feasibility of having a single transition model that slowly transitions from training on an objective timescale to training on a subjective timescale, as the memory formation goes on.

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

# A PRELIMINARIES

## A.1 ACTIVE INFERENCE

Active inference is a corollary of the free-energy principle applied to action (Friston et al., 2016; Friston, 2019; Sajid et al., 2019). In this framework, an agent embedded in an environment aims to do two things: (i) minimise surprisal from the observations of the environment under the agent's internal model of this environment, and (ii) perform actions so as to minimise the expected surprisal in the future. More formally, an agent is equipped with a generative model $p(o_t, s_t; \theta)$, where $o_t$ is the agent's observation at time $t$, $s_t$ is the hidden state of the environment, and $\theta$ denotes the parameters of the generative model. The agent's surprise at time $t$ is defined as the negative log-likelihood, $-\log p(o_t; \theta)$.

We can upper-bound this intractable expression using variational inference by introducing an approximate posterior distribution, $q(s_t)$, over $s_t$, such that:

$$-\log p(o_t; \theta) \leq \mathbb{E}_{q(s_t)} \left[ \log q(s_t) - \log p(o_t, s_t; \theta) \right] = \mathcal{F}, \tag{7}$$

where $\mathcal{F}$ is the *variational free energy*. The minimisation of this quantity realises objective (i) and is performed by optimising the parameters of the generative model, $\theta$. It is also equivalent to the maximisation of model evidence, which intuitively implies that the agent aims to perfect its generative model at explaining the sensory observations from the environment. To realise objective (ii), the agent must select actions that lead to the lowest *expected* surprise in the future, which can be calculated using the expected free energy (EFE), $G$:

$$G(\pi, \tau) = \mathbb{E}_{p(o_\tau | s_\tau)} \Big[ \underbrace{\mathbb{E}_{q(s_\tau | \pi)} \left[ \log q(s_\tau | \pi) - \log p(o_\tau, s_\tau | \pi) \right]}_{\text{variational free energy, } \mathcal{F}} \Big], \tag{8}$$

where $\tau > t$ and $\pi = \{a_t, a_{t+1}, ..., a_{\tau-1}\}$ is a sequence of actions (policy) between the present time $t$ and the future time $\tau$. The free-energy minimising system must, therefore, imagine the future observations given a policy and calculate the expected free energy conditioned on taking this policy. Then, actions that led to lower values of the EFE are chosen with higher probability, as opposed to actions that led to higher values of EFE, such that:

$$p(\pi) = \sigma\left(-\gamma G(\pi)\right), \tag{9}$$

where $G(\pi) = \sum_{\tau > t} G(\pi, \tau)$, $\gamma$ is the temperature parameter, $\sigma(\cdot)$ denotes a softmax function, and $t$ is the present timestep.

# B ARCHITECTURAL DETAILS AND TRAINING

## B.1 BASELINE IMPLEMENTATION

As mentioned, each component of the generative and inference models is parametrised by feed-forward neural networks (including fully-connected, convolutional and transpose-convolutional layers), whose architectural details can be found in Figure 6. The latent bottleneck of the autoencoder, $s$, was of size 10. The hyperparameters of the top-down attention mechanism were: $a = 2$, $b = 0.5$, $c = 0.1$, and $d = 5$, chosen to match those in Fountas et al. (2020a). Similarly, we restricted the action space to just 3 actions – forward, left, right. For testing, we optimised the MCTS parameters of the baseline agent, setting the exploration hyperparameter $c_{explore} = 0.1$ (see Eq.10), and performing 30 simulation loops, each with depth of 1. The networks were trained using separate optimisers for stability reasons. The habitual and transition networks are trained with a learning rate of 0.0001; the autoencoder's optimiser had a learning rate of 0.001. The batch size was set to 50 and the model was trained for 750k iterations under a green observational prior. All of the networks were implemented using Tensorflow v2.2 (Abadi et al., 2015). Tests were performed in Animal-AI v2.0.1 (Beyret et al., 2019).

Furthermore, following Fountas et al. (2020a), we define the MCTS upper confidence bound as,

$$U(s, a) = \tilde{G}(s, a) + c_{explore} \cdot Q_{\phi_a}(a|s) \cdot \frac{1}{N(s, a) + 1} \tag{10}$$

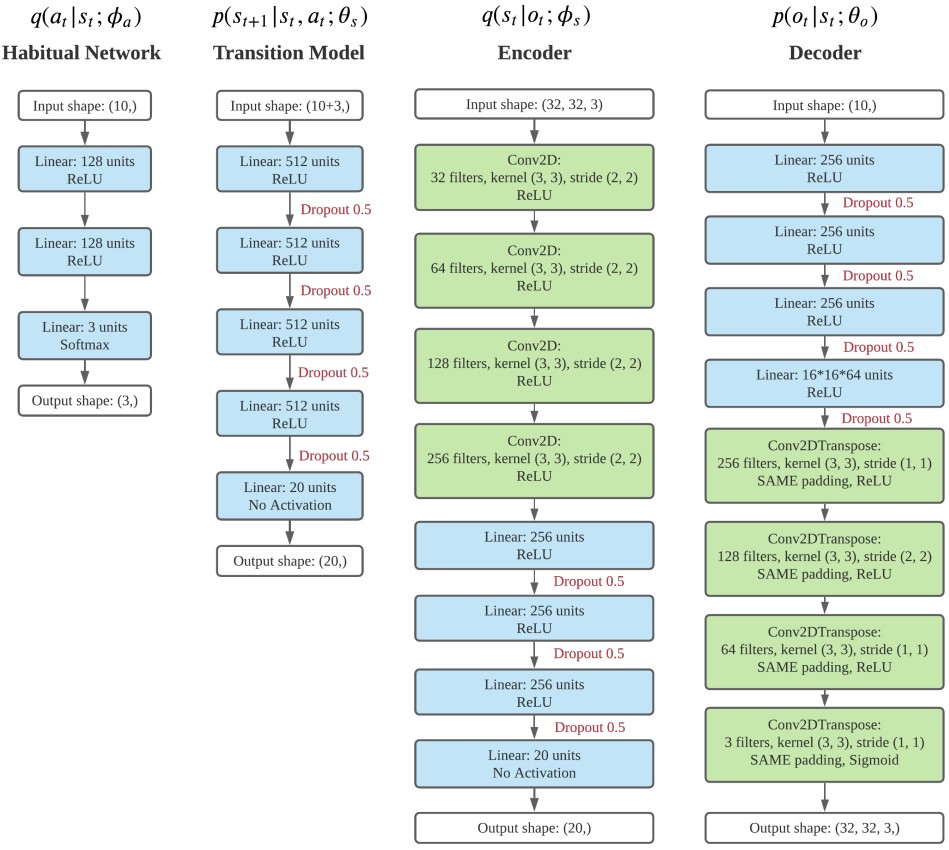

Figure 6: Implementation of the baseline system.

As discussed, each network was trained with its corresponding loss function, which are the constituent parts of the total variational free energy. In particular, the autoencoder was trained using Eqs. 1a and 1b, transition model using Eq. 1b, and habitual network using Eq. 1c.

Furthermore, following the training procedure from Fountas et al. (2020a), we stabilise the convergence of the autoencoder by modifying the loss function to:

$$
\begin{aligned}
\mathcal{L}_{\text{autoencoder}} = & -\mathbb{E}_{q(s_t)}\left[\log p(o_t|s_t;\theta_o)\right] + \gamma D_{\text{KL}}\left[q(s_t;\phi_s)||p(s_t|s_{t-1},a_{t-1};\theta_s)\right] \\
& + (1-\gamma)D_{\text{KL}}\left[q(s_t;\phi_s)||N(\mathbf{0},\boldsymbol{I})\right],
\end{aligned}
\tag{11}
$$

where $\gamma$ is a hyperparameter that gradually increases from 0 to 0.8 during training.

### B.2 PRIORITISED EXPERIENCE REPLAY

As part of the baseline system's training procedure, we utilise prioritised experience replay (PER) (Schaul et al., 2016) to mitigate the detrimental effects of on-line learning (which was used in the original paper by Fountas et al. (2020a)), and to encourage better object-centric representations.

In particular, on-line learning has three major issues associated with it. First, training is performed on correlated data points, which is generally considered to be detrimental for training neural networks (Schaul et al., 2016). Second, observations that are *rarely encountered* in an environment are discarded in on-line learning and are used for training *only* when visited again. These are likely to be the observations for which there is most room for improvement. Instead, the agent will often be training on already well-predicted transitions that it happens to visit often. Finally, an on-line learning agent is constrained by its current position in an environment to sample new data and, thus, has very limited control over the content of its training batches.

Furthermore, as mentioned in Section 4.1, in the Animal-AI environment rare observations are those that include objects; yet, objects are a *central* component of this environment – the only way to interact, get rewards, and importantly, the only means of minimising the free energy optimally. To encourage our agent to learn better object representations, we employ PER with the objective-timescale transition model free energy as the *priority metric*. As discussed, observations with higher values of this metric tend to constitute more complex scenes, which include objects – as the only source of complexity in the AAI. See Figure 7 for qualitative evidence of this trend. The use of PER resulted in a considerable improvement in the baseline's performance and better ability to reconstruct observations with objects (See Figure 8).

## B.3 STM IMPLEMENTATION

The STM introduces two additional components: STM habitual network and STM transition model. The habitual network was trained using the same training procedure as described in Appendix B.1. The transition model was trained on batch size 15 and a learning rate of $0.0005$. Each batch consisted of zero-padded S-sequences with length 50. We use a Masking layer to ignore zero-padded parts of the sequences in the computational graph. The training was stopped at 200k training iterations. For testing STM-MCTS in Section 5.1, we optimise the MCTS parameters, setting $c_{explore} = 0.1$, and performing 15 simulation loops, each with depth of 3. The threshold (objective-timescale transition model free energy), $\epsilon$, was manually set to 5 after inspection of the buffer and value distribution.

## B.4 ACTION HEURISTIC

To train the STM transition model in the Animal-AI environment, we implement a simple heuristic that is used to summarise a sequence of actions taken by the agent from one memory to reach the next one. A sequence of actions, $A = \{a_{\tau_1}, a_{\tau_1+1}, ... a_{\tau_1+(N-1)}\}$, takes the agent from a recorded memory $s_{\tau_1}$ to memory $s_{\tau_2}$, where the time between these states $\tau_2 - \tau_1 = N$, and $a \in \{a_{\text{forward}}, a_{\text{right}}, a_{\text{left}}\}$. We employ polar coordinates relative to the agent's initial position in Cartesian coordinates at time $\tau_1$ and perform iterative updates of its position after every action until the time-step of the next episodic memory, $\tau_2$, is reached. Given the agent's orientation in the environment, $\theta$, the next position of the agent is calculated using,

$$p_{t+1} = p_t + \begin{bmatrix} \sin\theta \\ \cos\theta \end{bmatrix}, \text{ where } p_t = \begin{bmatrix} 0 \\ 0 \end{bmatrix}_{t=\tau_1} \tag{12}$$

Finally, we retrieve angle $\phi$, which describes the direction in which the agent has travelled with respect to its initial position and orientation. This angle is used to decide on the action that summarises the trajectory using

$$a = \begin{cases} a_{\text{forward}} & |\phi| \leq 22.5° \\ a_{\text{right}} & 22.5° < \phi < 180° \\ a_{\text{left}} & -22.5° > \phi \geq -180°, \end{cases}$$

Although this heuristic provided satisfactory results, trajectory encoding is one of the most limiting parts of the STM and is a promising direction for further research.

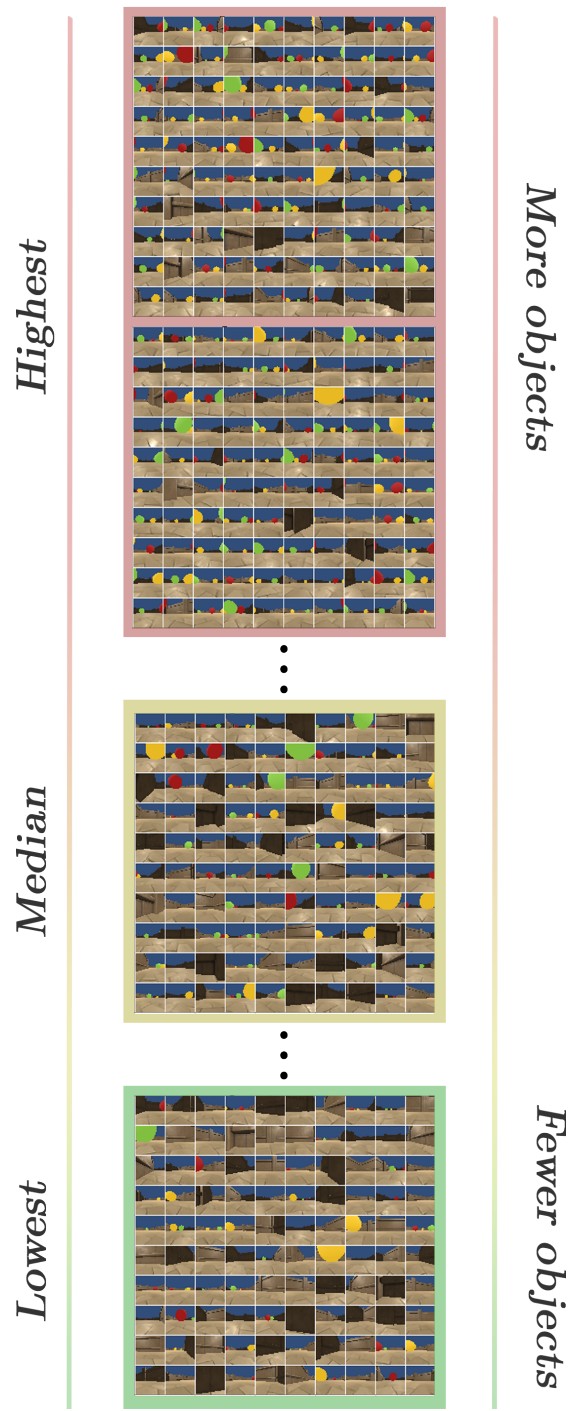

Figure 7: Observations sorted by their corresponding recorded value of the objective-timescale transition model free energy in descending order. States with higher values on average contain more objects, constituting more complex settings.

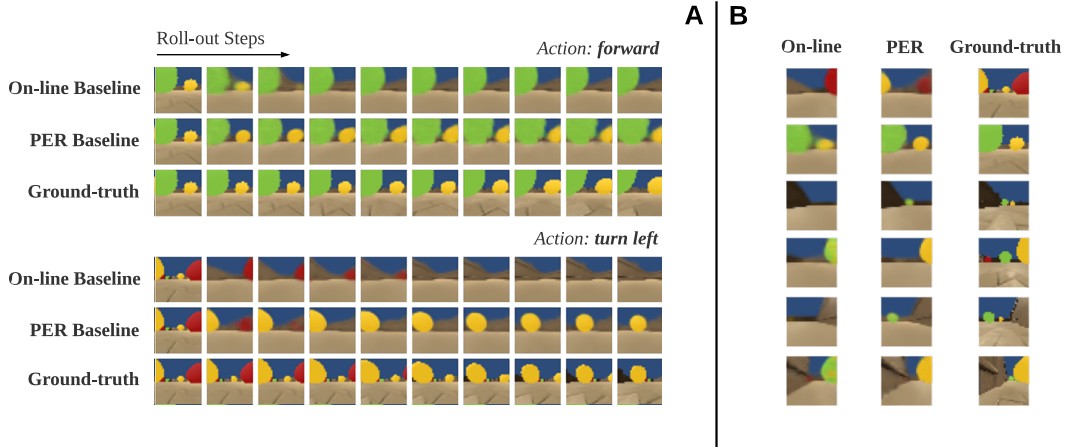

Figure 8: Agent trained with PER is better at generating observations with objects. **(A)** Prediction roll-outs showing that baseline trained with PER can better represent objects, thus preserving them in future predictions. **(B)** Ground-truth observations (third column) are passed through the autoencoder of the on-line and PER agents. As can be seen, observations are better reconstructed by the baseline system trained with PER.

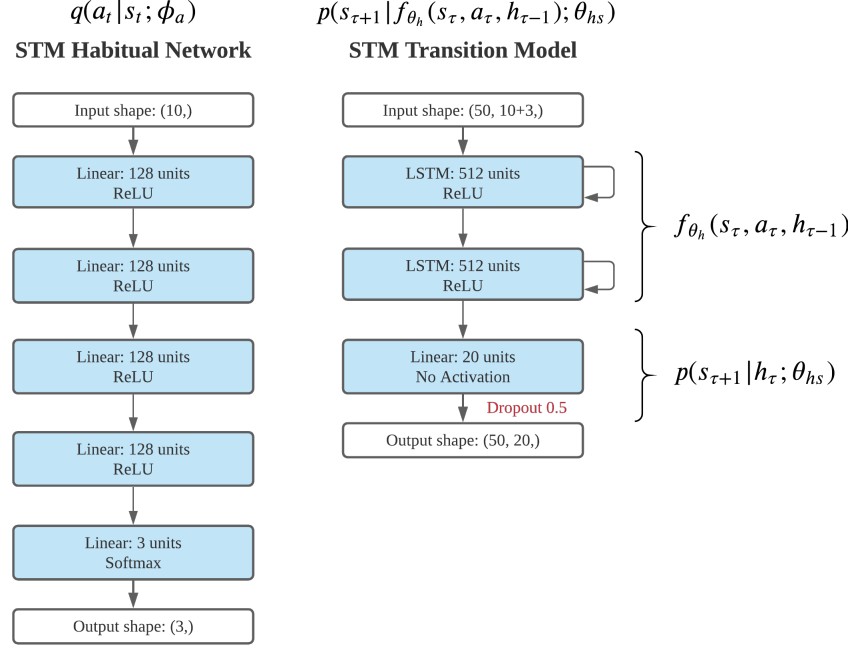

Figure 9: Implementation of STM components.

## C   ADDITIONAL RESULTS

We provide additional results of the STM prediction roll-outs:

a) Figure 10: random roll-outs generated by the system. These diverse roll-outs demonstrate that STM is able to: i) make correct *action-conditioned* predictions, ii) *speed up* its prediction timescale when objects are far away, iii) *slow down* the prediction timescale when objects are nearby.

b) Figure 11: STM consistently imagines objects coming into view. The observations produced by the model are entirely plausible given the path the agent is taking and the context it finds itself in. This indicates that STM does indeed produce semantically meaningful predictions. It is pertinent to note that the roll-outs comply with the physics of the environment, which is crucial, as it potentially refutes the hypothesis that these imagined objects were predicted at random.

c) Figure 12: shows the roll-outs produced by the objective-timescale model using *the same* starting states as in Figure 11. These roll-outs are in stark contrast to those produced by STM, exemplifying the baseline's inability to imagine objects that are not present in the initial frame.

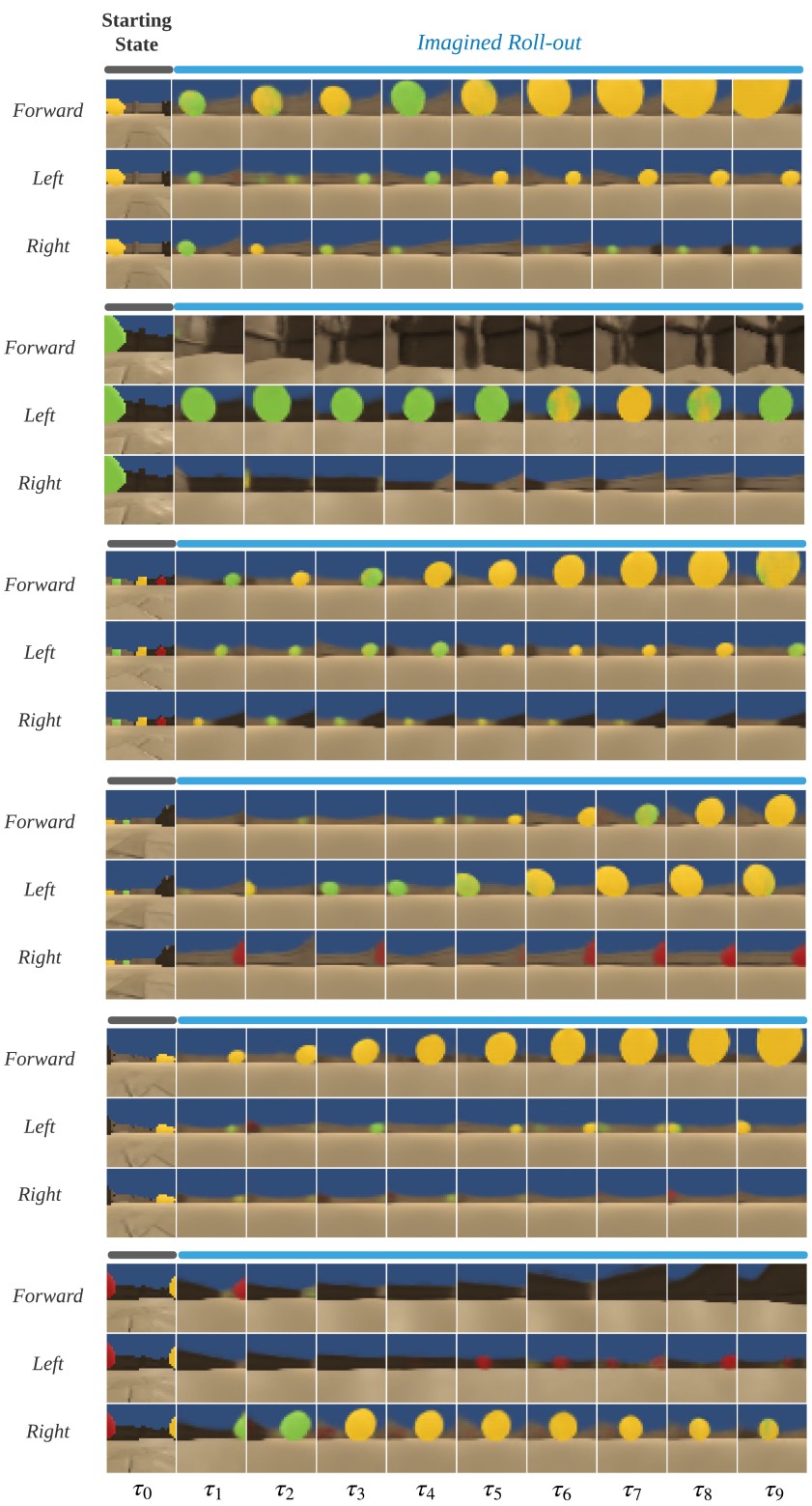

Figure 10: Random roll-outs generated with the STM transition model.

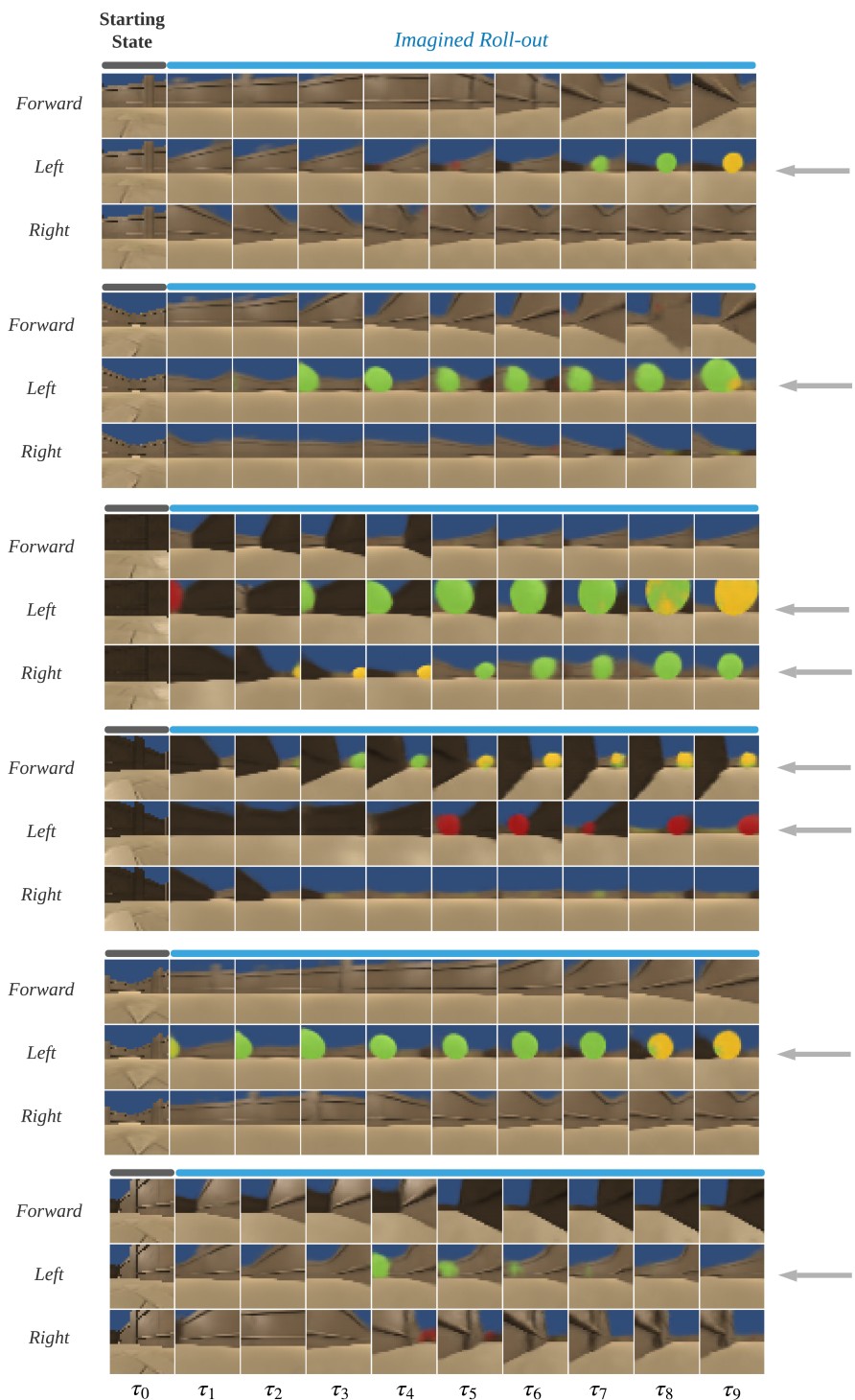

Figure 11: STM can imagine objects from 'uninteresting' states. Arrows indicate roll-outs with imagined objects.

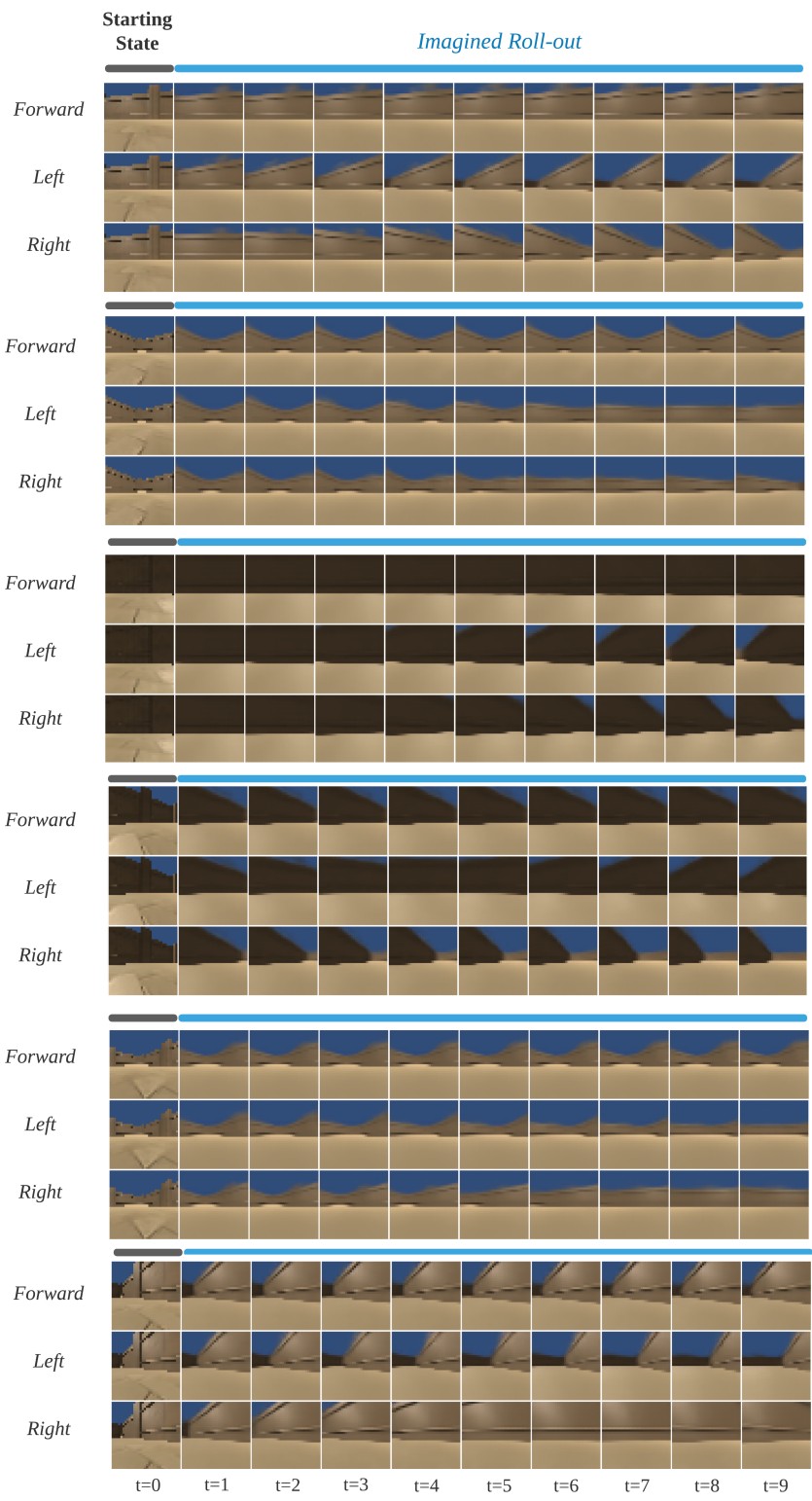

Figure 12: In contrast to STM, the objective-timescale transition model is not able to imagine objects, starting with the same initial observations as shown in Figure 11.

