# OpenReview forum: "Episodic Memory for Learning Subjective-Timescale Models"
_ICLR.cc/2021/Conference — Reject_

### Official Review · AnonReviewer3 · 2020-10-26
**Interesting problem and idea, but a little more work to be done**

**Rating:** 4
**Confidence:** 4

**Review:**

Summary:

Most model-based RL algorithms learn dynamics models that predicts the next timestep. However, because of model-bias, frequency of timesteps, and objective timescales, the dynamics models can accumulate errors and limited by timescales. The authors propose subjective-timescale model (STM) that instead of predicting the next timesteps they find the "surprising" subsequences of the trajectories and learn temporal-skipping dynamics models over them. The paper shows the improvement over single-step prediction baselines in a first-person navigation domain.

Pros:

The method aims to address a very important problem in model-based RL.

The idea of using variational free energy with model-based RL seems novel to me, and has not been widely explored.

The qualitative visualization (figure 4 and 5) provides a nice understanding of what the method is doing as well as what it is capable of in first-person navigation.

Cons:

-- Methods --

The main sections do not contain sufficient information regarding how the actions are obtained from learned STM models. I find one paragraph in section 3 in sufficient. The experimental sections also do not mention how MCTS and MPC baselines differ. Please clarify. Also, how do you recover low-level action sequences from the aggregated actions after MCTS? I do not find the answer from the paper.

The keyframe selection method requires more justification. It is unclear how using the KL divergence for measuring surprise will improve over model's prediction error studied in previous work.
It would be amazing to provide a theoretical justification for the heurstics toward, e.g., saying something about the end task performance, if possible. How sensitive is it to the KL threshold? Please provide this study.

The action sequence aggregation is domain specific which seems a bit unfair to compare against the baselines which **do not** have access to the same information. There should be more baselines or ablation studies to disentangle the improvement of the method from this domain-specific assumption.

The paper uses different indexing styles which make the method more confusing than it should have been. Please choose one between indexing tau or arithematics on tau.

-- Experiments --

As metioned briefly before, more baselines or ablation will be critical to judge the importance of the proposed model? What about compare against other sliency approaches such as prediction error for memory accumulation? Also, it would be helpful to have more than 1 environment to show the genrality of the approach.

The experimental results do not provide enough information to understand what tasks can be solved and what cannot be solved in Animal AI environment. Can the we provide the success rates and categorized by difficulty levels? These information will be helpful in understanding what STM can and cannot do. Perhaps, having a link to some videos will also help.

Figure 3 shows that the reward is going up; how far can the rward go? It is still increasing.

In figure 3, how is the reward computed?

In figure 5, is there a groundturth trajectory comparison?


-- Others --

Personally, I would appreciate more background and intuition on each term in the variational free energy formula.

Conclusion:

Again, I believe that this work is addressing a very important question with an interesting idea, but it may require a little bit more work to make the case. I appreciate the authors thinking about this problem, and hope the authors are encouraged to continue their work.

---

> ### Author Response · Authors · 2020-11-16
> **In response to AnonReviewer3 (part 1)**
>
> Dear AnonReviewer3,
>
> Thank you for your review. We are happy to address the concerns you outlined above.
>
> **Q1**: *“The main sections do not contain sufficient information regarding how the actions are obtained from learned STM models.”* + *“how MCTS and MPC baselines differ”*
> **A**: We understand your concern here. We will extend these explanations in the next version of the manuscript.
>
> **Q2**: *“how do you recover low-level action sequences”*
> **A**: Our model does not involve a hierarchical approach for retrieving action sequences. While we acknowledge that this may seem as a disadvantage (since there is no one-to-one correspondence in the predicted future states and the ground-truth states), the retrieval of low-level sequences of actions is not necessary to observe a visible improvement in the system’s performance. As mentioned in another response, we are now working on more sophisticated methods for retrieving useful long-term action sequences and hope this will improve the results further.
>
> **Q3**: *“It is unclear how using the KL divergence for measuring surprise will improve over model's prediction error studied in previous work.”*
> **A**: Could you please let us know some of the exact references you are referring to so we can better investigate this point. As we understand it, KL divergence (in this case, variational free energy of the dynamics model) can also be considered to be the prediction error produced by the probabilistic transition dynamics model. Within active inference, it is used to quantify a measure of belief updating and thus, loosely speaking, quantifies the prediction error, as well. As we mentioned in the paper, empirical evidence from neuroscience suggests that prediction error and event saliency is one of the decisive factors on whether an episodic memory will be formed.
>
> **Q4**: *“How sensitive is it to the KL threshold?”*
> **A**: Thank you for raising this point, we agree with you that more should be said about the threshold value choice. Currently, the threshold was treated as a hyperparameter of the model and was manually chosen by inspecting the distribution of transition model surprise values.
>
> **Q5**: *“unfair to compare against the baselines which do not have access to the same information.”*
> **A**: We are not entirely sure what you mean by this. Both the baseline and the STM agents have access to exactly the same amount of information. The action sequence aggregation is formed out of the actions performed by the agent, which the objective timescale agent also has access to (though of course stored in the objective timescale).
>
> **Q6**: *“it would be helpful to have more than 1 environment to show the generality of the approach.”*
> **A**: We certainly agree with this. However, we believe that the presented results do show that the idea is sound and strongly suggest that it has potential in other domains. We are currently working on further experiments across more benchmarks/environments.
>
> To be continued in the next post...

---

> > ### Author Response · Authors · 2020-11-16
> > **In response to AnonReviewer3 (part 2)**
> >
> > **Q7**: *“The experimental results do not provide enough information to understand what tasks can be solved and what cannot be solved in Animal AI environment”*
> > **A**: We have tested our agents on a number of tasks from the official 2019 Animal-AI Competition; however, we decided not to report these, as we believe it would distract the reader from the main theme of the paper. It is also important to mention the apparent difficulty of testing active inference agents on reward-based tasks (which is characteristic to the testing of RL agents) given that the agent cannot learn from the reward in the same way. We broadly follow the discussion in Fountas et al. Neurips 2020 [1], which further expands on this point.
> >
> > **Q8**: *“the reward is going up; how far can the reward go?”*
> > **A**: The figure shows the cumulative reward from 100,000 randomly-generated environments, rather than average reward per episode.
> >
> > **Q9**: *“how is the reward computed?”*
> > **A**: The reward is provided by the environment: ~5 for reaching a yellow/green sphere, ~(-5) for reaching a red sphere, and a constant negative reward of ~(-0.01) at every timestep.
> >
> > **Q10**: *“is there a ground truth trajectory comparison?”*
> > **A**: We believe it is not informative for this particular figure, as the most interesting beneficial property of the STM agent is that it predicts imagined salient events (i.e. not necessarily corresponding to the ground truth - which cannot be predicted in this case), and also because the two sequences play out over different timescales. The figure shows the difference between our STM agent (which is able to imagine salient/rare events in its roll-outs) and the baseline agent (which simply predicts the most likely next frame). The ground truth is given in the other figures, however (4, 8).
> >
> > **Q11**: *"I would appreciate more background and intuition on each term in the variational free energy formula."*
> > **A**: Thank you for the suggestion – this seems to be a common wish. We will extend the appendix explanation of active inference.
> >
> > Thank you again for your review.
> >
> > Best regards.
> >
> > [1] https://proceedings.neurips.cc/paper/2020/hash/865dfbde8a344b44095495f3591f7407-Abstract.html

---

### Official Review · AnonReviewer4 · 2020-10-28
**Subjective time perception is a nice motivation. Execution could be significantly improved.**

**Rating:** 4
**Confidence:** 3

**Review:**

Summary: The authors propose to train a model not on the objective time-scale of a sequence of frames, but on the subjective time-scale dictated by how surprising events are (where surprise here is defined as being above a certain energy free threshold). The trained action-conditioned model learns to slow down time for complex scenes, and fast forward when things are easily predicted.

The overall topic is an important one, most model based methods suffer from accumulating errors.
The introduction is well written and offers a strong motivation for the rest of the paper. I like the explanation in terms of a distinction between objective and subjective perception of time and events.

There are lots and lots of references to work by Friston et al., but I am not sure I see the connection with active inference as being that strong or even necessary.

Regarding the part “Furthermore, for long-term predictions STM systematically performs temporal jumps (skipping intermediary steps), thus providing more informative future predictions and reducing the detrimental effects of one-step prediction error accumulation.” There are two potentially relevant references (Neitz et al NeurIPS  2018), and (Darajaman et al, ICML 2019), as both learn models (not necessarily action-conditioned though) that can skip an adaptive number of steps into the future, with similar consequences (i.e. preventing error accumulation and increasing rollout speed).

P4: “The habitual network acts as a model-free component of the system, learning to map inferred states directly to actions”. Shouldn’t $q(a_t; φ_a)$  be a function of $s$ as well?

P6: The angle heuristic might deserve a more in-depth discussion. What if the agent goes in a circle? Or does a U turn? Or a complex sequence of movements in a maze?
Since the goal of STMs is (at least in part) to reduce progressively the length of S-sequences (such that they start spanning longer and longer horizons), the actions that bridge two episodic memory need to be summarised in a way that is expressive enough.
Could one think of applying the same STM principle that is already applied to states, to actions as well?

“Importantly, [the] function $f_{\theta_{s}}$ is deterministic and serves only to encode information about preceding…”. Where does $f_{\theta_{s}}$ appear in this context? I can’t see it anywhere between eq. 4 and 6

This AAI environment is not so established. Perhaps the author could add (even in the appendix) a top view of a typical setting that corresponds to one episode?
Also, it appears that the setting chosen by the authors does not reflect the characteristics that they planned to showcase with their method. Given the presence of one sphere per colour and a relatively small environment, how can we appreciate that the agent learns to do planning over long horizons?
Perhaps an environment with longer horizons would be a more adequate testbed?

I very much liked that the authors showed the additional results in Appendix C, I think they are extremely important for the paper. However, I disagree with the claims made in the text, as it appears that they are not substantiated by the figures they reference.
- “Figure 10: random roll-outs generated by the system. These diverse roll-outs demonstrate that STM is able to: i) make correct action-conditioned predictions, ii) speed up its prediction timescale when objects are far away, iii) slow down the prediction timescale when objects are nearby”
If I interpret Figure 10 correctly, it seems to suggest that the imagined roll-outs are not very consistent, as in almost every single case the color of the spheres changes from green to yellow and vice-versa during the rollout.
- Figure 11: several transitions appear not to be realistic, objects appear out of nowhere instead of smoothly while turning.
- Figure 12: while it is true that the baseline model does not generate any sphere, it appears to me that the physics is significantly more consistent than with STM.

The last sentence in the conclusion seems to suggest that the model is not progressively expanding its horizon the more it trains. How come?

Overall, my impression is that the paper has a very interesting motivation, but the execution could be significantly improved. The results are not so convincing, due to:
1) figures (10-11-12) that do not soundly corroborate the claims,
2) evaluation setting that does not allow to really test the claim (i.e. not enough opportunities for increasingly longer horizons as the model improves)
3) the lack of established baselines and benchmarks
4) the lack of an algorithm box to present the method

Minor:
- Simplify language where possible (utilised -> used; are capable of -> can; etc.)
- Several figures (4,5, 8) are corrupted using preview on a Mac (not sure if it’s just my computer). I could see them correctly by using Chrome to open the PDF.

---

> ### Author Response · Authors · 2020-11-16
> **In response AnonReviewer4 (part 1)**
>
> Dear AnonReviewer4,
>
> Thank you for your comments and feedback. We would love to address the issues you bring up in your review.
>
> **Q1**: *“I am not sure I see the connection with active inference as being that strong or even necessary”*
> **A**: Active inference is a model-based cognitive framework that we chose mainly for the reasons of biological plausibility and its intrinsic Bayesian nature. This, as well, is consistent with the Bayesian predictive processing model for time perception used in Fountas et al. (2020) [1]. Regardless, we would kindly want to ask you to clarify a little more on this concern, particularly with respect to the connection with active inference not being “that strong”.
>
> **Q2**: *“There are two potentially relevant references [...]”*
> **A**: Thank you for these! However, we were unable to locate the Darajaman et al. in the proceedings of ICML 2019. Could you please link the paper you are referencing?
>
> **Q3**: *“Shouldn’t* $q(a_t; \phi_a)$ *be a function of s as well?”*
> **A**: You are correct in saying that it is also a function of s. Here, however, we follow standard notation from variational inference which drops the conditioning, when denoting the approximate posterior. We chose to stick with the conventions in the literature, but agree that writing it out in full may be more clear.
>
> **Q4**: *“The angle heuristic might deserve a more in-depth discussion. What if the agent goes in a circle? Or does a U turn? Or a complex sequence of movements in a maze?”*
> **A**: Absolutely! We certainly agree with you that the angle heuristic deserves more attention, and we plan to address it more concretely in our future work. The purpose of this paper, however, was to showcase the usefulness of defining a subjective timescale as the top priority. Therefore, we chose a very simple heuristic that was necessary to provide the agent with information to learn action-conditioned predictions. This is not to say that they are perfect and we would expect them to get worse in more complex configurations of the environment; however, we believe that this heuristic was enough to demonstrate the effectiveness of subjective-timescale models.
>
> **Q5**: *“Could one think of applying the same STM principle that is already applied to states, to actions as well?”*
> **A**: Could you please clarify the suggestion as there are many ways this might be possible, and it's unclear to us exactly how you are imagining this to work.
>
> **Q6**: *“Where does $f_{\theta_s}$ appear in this context?”*
> **A**: Thank you for pointing it out – it is a typo and should be $f_{\theta_h}$, instead.
>
> **Q7**: *“Perhaps the author could add (even in the appendix) a top view of a typical setting that corresponds to one episode?”*
> **A**: Definitely, thank you for the suggestion.
>
> **Q8**: *“the setting chosen by the authors does not reflect the characteristics that they planned to showcase with their method… . [...] how can we appreciate that the agent learns to do planning over long horizons?”*
> **A**: Although it may seem that the environment is small, we argue that the most important bit is the underlying temporal dynamics – e.g. how much does an agent progress forward given a forward action? In AAI, it is very slow, requiring the agent to take hundreds of steps (>500) to go to the opposite side of the sandbox. Furthermore, sparse rewards were chosen deliberately to encourage both long- and short-term planning. This becomes even more challenging, as our agents were only allowed to take 500 steps, after which the environment would terminate and the next configuration would be chosen. Therefore, we believe that the chosen set-up, on the contrary, helps with testing our agents for both short- and long-term planning abilities. Now that the idea has been shown to be successful we plan to address other benchmarks/environments that are likely to display more traditional long-term planning behaviour.
>
> To be continued in the next post...
>
> [1] https://www.biorxiv.org/content/10.1101/2020.02.17.953133v1

---

> > ### Author Response · Authors · 2020-11-16
> > **In response to AnonReviewer4 (part 2)**
> >
> > **Q9**: *“[Figure 10] … imagined roll-outs are not very consistent, as in almost every single case the color of the spheres changes from green to yellow”*
> > **A**: The observed interchange between the yellow and green colours was attributed to the fact that the STM was only trained for 250k iterations in the presence of close clustering of yellow and green colour in the latent space by the regularised VAE. We believe this issue can be addressed by training the dynamics model for longer, as this problem was not observed in the objective-timescale agent, which shared the same weights for its VAE and which had its dynamics model trained for about 750k iterations. With Figure 10, however, we wanted to draw your attention to the apparent variability in the temporal extent of the agent’s predictions, as well the consistency in the agent’s action-conditioned predictions. For instance, Roll-out #1 shows an initially rapid forward approach to the sphere, which later slows down, while Roll-out #2 correctly predicts observing a green sphere when turning left. We encourage the readers of this paper to think of the subjective-timescale model as not strictly speaking a conventional dynamics model that is attempting to model the true physics or the most likely next state given an action. Rather, the STM will guide the agent by means of predicting salient events over varying time intervals. Nevertheless, we also agree that our results could be made more consistent and visually convincing, and we thank you for pointing to some of these issues.
> >
> > **Q10**: *“[Figure 11] … objects appear out of nowhere instead of smoothly while turning”*
> > **A**: We believe this is consistent with what we would expect the STM to predict. This is because episodic memories contain ‘surprising’ events, which we categorise into two main categories (Section 4.1, Paragraph 3) – epistemic and model-imperfection surprise. As a result of the former, the episodic memory buffer will contain events like spheres appearing in the frame of view when the agent turns. Furthermore, these events do not need to happen ‘smoothly’, because the surprise threshold is fixed at a specific value, below which the agent will not record a memory. We see it as a positive feature of the model in that it skips the less surprising smooth introduction of new objects at the edges.
> >
> > **Q11**: *“[Figure 12] … it appears to me that the physics is significantly more consistent than with STM”*
> > **A**: While it is true that the objective-timescale model is better at modelling some parts of the physics of the environment, in this paper we argue that that is not what the internal model must necessarily be good at, and can actually be its disadvantage (no imagination-driven exploration, more prone to get stuck in a sub-optimal state). Instead, we train the STM to predict future salient events, driving the agent to places where they are more likely to occur. Nevertheless, STM retains the ability to correctly predict in the short-term, which does not compromise the agent’s ability to get the reward (or avoid a negative reward) when it is close to one.
> >
> > **Q12**: *“The last sentence in the conclusion seems to suggest that the model is not progressively expanding its horizon the more it trains. How come?”*
> > **A**: Great question. This is because the objective-timescale transition model used to collect memories is pre-trained and frozen. We plan to explore using a single transition model in our future work, for which we would expect this behaviour.
> >
> > **Q13**: *“lack of established baselines and benchmarks”*
> > **A**: Please refer to our answer in Q8.
> >
> > **Q14**: *“an algorithm box to present the method”*
> > **A**: Thank you for the suggestion – we will add this.
> >
> > Thank you once again for your thoughtful review.
> >
> > Best regards.

---

> > > ### Comment · AnonReviewer4 · 2020-11-22
> > > **Reply**
> > >
> > > **Q1**: _we would kindly want to ask you to clarify a little more on this concern, particularly with respect to the connection with active inference not being “that strong”_ : There are more than 8 references to work by Friston et al., but it's unclear to me how relevant all these references are.
> > >
> > > **Q2**: _We were unable to locate the Darajaman et al. in the proceedings of ICML 2019. Could you please link the paper you are referencing?_ my mistake, the second reference was supposed to be Jayaraman et al, ICLR 2019 (TIME-AGNOSTIC PREDICTION: PREDICTING PREDICTABLE VIDEO FRAMES).
> > >
> > > **Q5**: Mine was just a vague suggestion, not meant to be addressed in this work. What I mean is, can the agent learn a subjective scale for actions, somehow similar to how it should do for states? Keeping the actions fine-grained might make most of the advantages of the subjective time-scale for states disappear. The heuristic you introduce helps to fix the issue, but it seems too strong to be meaningful (e.g. no U turns, sequences of turns, etc.).
> > >
> > > **Q8**: I am still not convinced that this environment is sufficient to showcase long horizons plans. Especially with the very coarse action heuristic in place I don't see how the agent can plan long-term, if it doesn't even know what actions it executed. This is consistent with the fact that the simulated plans tend to be not realistic and of worse quality than the standard planner.
> > > All in all, I find this experiment with Animal AI not convincing, and it's a pity because the idea is interesting and deserves a solid evaluation.
> > >
> > > **Q9**: _We believe this issue can be addressed by training the dynamics model for longer_ This needs to be shown empirically, given the analysis in the paper right now I don't see this would be a given.

---

### Official Review · AnonReviewer1 · 2020-10-28
**Important problem and interesting agent, but needs more in-depth analysis.**

**Rating:** 5
**Confidence:** 3

**Review:**

In this paper, the authors describe a variable-timescale prediction model for planning in the context of a deep active inference agent. They show that this agent outperforms a baseline in a scavenging task in a 3D first person environment. They show example rollouts of the baseline and variable-timescale models.

Until reading this paper I wasn't familiar with deep active inference agents, which apparently enable the extension of free energy methods to more complex settings. It seems like an intriguing alternative to deep RL. It's not clear to me whether it has the scaling potential that has been demonstrated for deep RL. Although the experiments reported here are larger scale than experiments with active inference systems prior to Fountas et al. (2020), they seem to be quite simple in comparison to tasks on which deep RL agents excel (e.g. typical Vizdoom and DMLab tasks). I don't mean this as a criticism, but instead as a question mark: this difference in demonstrated scale might simply be due to the massive difference in resources that have been devoted to deep RL agents vs active inference agents.

As a result, I tried to evaluate the paper on its merits in the context of active inference systems, instead of via comparison with deep RL. As such, I'd be happy with a convincing demonstration that (1) the variable timescale model outperforms a strong time-locked model in the scavenging task and (2) that its performance is due to the selection of useful frames for planning that MCTS then uses effectively.

Figure 3 seems to answer (1) in the affirmative. I'm basically trusting the authors that theirs is a reasonably strong baseline, since I don't have experience training active inference agents or with this scavenging environment. This will be reflected in my confidence score.

However, (2) is not substantiated well by the paper's analysis section. The only evidence for this is in the form of two pairs of example rollouts for the time-locked and variable-time models. Are these cherry picked examples, or are they actually reflective of general trends? I'd be much more comfortable if the authors supplied some aggregate results to substantiate this claim:

"As a result, our agent consistently predicts farther into the future in the absence of any nearby objects, and slows its timescale, predicting at finer temporal rate, when the objects are close."

I'd really need aggregate results demonstrating that the S-sequences are summarizing long trajectories in a sensible way over a large set over episodes, to feel confident that the system is providing the benefits its purported to. Even better would be to analyze the MCTS search trees to show that the search trajectories over S-seqences have desirable properties.

Perhaps more concerningly, the authors say that "As a result, the STM agent is less prone to get stuck in a sub-optimal state, which was commonly observed in the baseline system, and is more inclined to explore the environment beyond its current position". But, as far as I can tell, there's no evidence in the main paper for this.

One concrete concern is that the time-locked agent fails simply because it's rollouts aren't long enough to find the rewarding object. Maybe it would be sufficient to simply randomly drop out timesteps from the trajectory to reach the STM-MCTS performance level. The current analyses (and baseline results) don't rule out this hypothesis.

If the authors can provide stronger evidence on these points, I'd be very happy to increase my rating.

---

> ### Author Response · Authors · 2020-11-16
> **In response to AnonReviewer1**
>
> Dear AnonReviewer1,
>
> Thank you for your thoughtful feedback – you raise some important points, which we would be happy to address.
>
> **Q1**: *“[...] this difference in demonstrated scale might simply be due to the massive difference in resources that have been devoted to deep RL agents vs active inference agents.”*
>
> **A**: We certainly agree with this statement and believe that deep active inference research could greatly benefit from more computational resources and efforts of scaling it further.
>
>
> **Q2**: *“I'm basically trusting the authors that theirs is a reasonably strong baseline”*
>
> **A**: Our baseline is a deep active inference agent devised by Fountas et al. (NeurIPS 2020), which was compared to several model-free RL agents, such as DQN, A2C, and PPO2. We believe it can be considered the current state of the art for deep active inference agents. Furthermore, the parameters of this baseline were tuned to yield better performance. We would also like to stress that the only difference between the baseline and our STM agent is the transition model. As such, we can much more confidently attribute the improvement in the performance to the subjective-timescale model.
>
>
> Moving on, you made several points about being uncertain whether the STM agent’s better performance is indeed due to the subjective-timescale model. Accordingly, we plan to address these in the next version of the manuscript and we thank you for pointing it out. Nevertheless, in response to your concern, we again emphasise that the baseline and STM agent share the weights of all the networks, except for the transition dynamics model. This was done deliberately to address the exact point that you raise.
>
>
> **Q3**: *“Are these cherry picked examples, or are they actually reflective of general trends?”*
>
> **A**: These are indeed the general trends that we observed by performing random roll-outs in a variety of different settings. For more examples of random roll-outs, please see Figures 10 and 11 in the Appendix.
>
> Nevertheless, we believe your concerns about the analysis of the results, such as demonstrating *“aggregate results”* and *“analyz[ing] the MCTS search trees”*, are well-justified. We are currently working towards creating more convincing quantitative and qualitative metrics, by which these improvements could be judged.
>
>
> **Q4**: *“[With regards to baseline agents getting stuck in sub-optimal states],  there's no evidence in the main paper for this.”*
>
> **A**: This observation was based on numerous runs we performed and analysed qualitatively. We can find ways to support this observation more quantitatively in the next version.
>
>
> **Q5**: Further, you mention that the hypothesis that *“[...] the time-locked agent fails simply because it's rollouts aren't long enough to find the rewarding object”* is not ruled out by the analysis.
>
> **A**: There are several points to address here:
> (1) We did experiment with both longer and shorter roll-outs for the time-locked agent, and found that performing longer roll-outs does not result in a better performance – likely related to the problem of error accumulation for one-step prediction models (as can be observed in Figure 4).
> (2) Time-locked models can indeed yield roll-outs that are not long enough; however, it is exactly one of the issues that our STM model can effectively address – but do so without resorting to any explicit mechanisms of varying the temporal extent of predictions.
> (3) Longer roll-outs result in higher computational complexity of the planning process – something that is addressed via the subjective-timescale modelling.
> (4) As shown in Figures 5 and 11, STM agents can additionally imagine objects (affordances that would allow for optimal minimisation of the free energy), which is in stark contrast to the objective-timescale agent in Figures 5 and 12. We argue that this systematic characteristic allows the STM agent to have richer information about its environment (potential affordances), encouraging imagination-driven exploration, while the time-locked agent is deprived of such ability.
>
>
> We again thank you for your useful feedback, and we will continue working on improving our paper in the meantime.
>
> Best regards.

---

> > ### Comment · AnonReviewer1 · 2020-11-19
> > **Reply to Authors**
> >
> > The authors' response to Q2 makes sense overall and is convincing, thanks. In a bit more depth:
> >
> > < Our baseline is a deep active inference agent devised by Fountas et al. (NeurIPS 2020), which was compared to several model-free RL agents, such as DQN, A2C, and PPO2. We believe it can be considered the current state of the art for deep active inference agents. >
> >
> > That's good to know. If the baseline performs similarly to those deep RL agents, specifically on this scavenging task, then knowing about that result would help me as reader with a deep RL background to situate the results of the present paper. The strongest evidence might be e.g. a plot showing DQN, A2C, PPO, and the free energy baseline performing similarly, then STM outperforming all of them. That might be a lot of work, so if there are scavenging tasks in Fountas et al 2020 on which the baseline performs favorably in comparison to DQN etc, then mentioning that in the present paper would go a long way.
> >
> > < We again emphasise that the baseline and STM agent share the weights of all the networks, except for the transition dynamics model. >
> >
> > I must have missed this when reading through. If this rules out alternative explanations for STM's improved performance, it might be worth unpacking that logic for the reader. As a sidenote, I'd have thought that sharing parameters like this would inhibit maximal performance for both architectures.
> >
> > Answers to Q3-Q5 also make sense, I'll look forward to the next version of the manuscript. One point in more detail:
> >
> > < We did experiment with both longer and shorter roll-outs for the time-locked agent, and found that performing longer roll-outs does not result in a better performance – likely related to the problem of error accumulation for one-step prediction models (as can be observed in Figure 4). >
> >
> > That's good to know. I think it would be convincing if the paper showed that: specifically, if you roll out the time-locked model to the longest environment timestep STM predicts, you get much worse performance. It might seem obvious given the assumption (which I think the authors hold from experience) that the time-locked model suffers severely from accumulating errors. But, for the uninitiated reader, something like this could serve as a quick sanity check and convincing demonstration of STM's usefulness.

---

### Decision · Program_Chairs · 2021-01-07
**Final Decision**

**Decision:**

Reject

**Comment:**

This paper uses a free-energy formulation to develop an approach to learning "jumpy" transition models, which predict surprising future states. This transition model is used in combination with MCTS and applied to a scavenging task in the Animal AI Olympics, outperforming two baselines.

While the reviewers praised the importance of the problem tackled, and the novelty of using a free energy approach, there was a general sense amongst the reviewers that the paper wasn't totally clear (especially for an RL audience). R1 also felt that some of the claims of the paper weren't sufficiently evaluated enough, and several reviewers indicated that they felt the baselines were insufficient (or, at a minimum, not described in enough detail to evaluate whether they were sufficient). Given these points, I feel the paper is not quite ready for publication at ICLR. I encourage the authors to flesh out their analysis a bit more, better describe the baselines (and possibly compare to other existing approaches as mentioned by R4), and overall to frame the paper a bit more for the RL community.

One additional reference the authors may be interested in: Gregor et al (2018). Temporal difference variational auto-encoder.